# Cooperation or Competition: Avoiding Player Domination for Multi-target Robustness by Adaptive Budgets

**Yimu Wang**
David R. Cheriton School of Computer Science
University of Waterloo
Waterloo, Canada
yimu.wang@uwaterloo.ca

**Dinghuai Zhang**
Mila, University of Montreal
Montreal, Canada
dinghuai.zhang@mila.quebec

**Yihan Wu**
University of Pittsburgh
Pittsburgh, United States
yiw154@pitt.edu

**Heng Huang**
University of Pittsburgh
Pittsburgh, United States
henghuanghh@gmail.com

**Hongyang Zhang**
David R. Cheriton School of Computer Science
University of Waterloo
Waterloo, Canada
hongyang.zhang@uwaterloo.ca

## Abstract

Despite incredible advances, deep learning has been shown to be susceptible to adversarial attacks. Numerous approaches were proposed to train robust networks both empirically and certifiably. However, most of them defend against only a single type of attack, while recent work steps forward at defending against multiple attacks. In this paper, to understand multi-target robustness, we view this problem as a bargaining game in which different players (adversaries) negotiate to reach an agreement on a joint direction of parameter updating. We identify a phenomenon named *player domination* in the bargaining game, and show that with this phenomenon, some of the existing max-based approaches such as MAX and MSD do not converge. Based on our theoretical results, we design a novel framework that adjusts the budgets of different adversaries to avoid player domination. Experiments on two benchmarks show that employing the proposed framework to the existing approaches significantly advances multi-target robustness.

## 1 Introduction

Machine learning models have been shown to be susceptible to adversarial examples (Szegedy et al., 2014), where human-imperceptible perturbations added to a clean example might arbitrarily change the output of machine learning models. The adversarial examples are generated by maximizing the loss within a small perturbation region around a clean example, *e.g.*, $\ell_\infty$, $\ell_1$ and $\ell_2$ balls. On the other hand, numerous heuristic defenses have been proposed to be robust against adversarial examples, *e.g.*, distillation (Papernot et al., 2016), logit-pairing (Kannan et al., 2018) and adversarial training (Madry et al., 2018).

2022 Trustworthy and Socially Responsible Machine Learning (TSRML 2022) co-located with NeurIPS 2022.

However, most of the existing defenses are only robust against one type of attacks (Madry et al., 2018; Raghunathan et al., 2022; Wong & Kolter, 2018; Engstrom et al., 2018), while they fail to defend against other adversaries. For example, existing work (Kang et al., 2019b; Maini et al., 2020) showed that robustness in the $\ell_p$ threat model does not necessarily generalize to other $\ell_q$ threat models. However, for the sake of the safety of machine learning systems, one should target robustness against multiple adversaries simultaneously (Croce & Hein, 2020a).

Recently, various methods (Schott et al., 2019; Tramer & Boneh, 2019; Maini et al., 2020) have been proposed to address this problem. Multi-target adversarial training, which targets defending against multiple adversarial perturbations, has attracted significant attention: a variational autoencoder-based model (Schott et al., 2019) learns a classifier robust to multiple perturbations; after that, MAX and AVG strategies, which aggregate different adversaries for adversarial training against multiple threat models, have been shown to enjoy improved performance (Tramer & Boneh, 2019). To further advance the robustness against multiple adversaries, Maini et al. (2020) achieved better performance than MAX and AVG by taking the worst-case over all steepest descent directions. These methods follow a general scheme as the (single-target) adversarial training.

This general scheme for multi-target adversarial training can be seen as an implementation of a cooperative bargaining game (Thomson, 1994). In this game, different parties have to decide how to maximize the surplus they jointly get. In the multi-target adversarial training, we view each party as an adversary, and they negotiate to reach an agreed gradient direction that maximizes their utility function. We can then utilize the results from a game-theoretic perspective to analyze the multi-target adversarial training.

Inspired by the bargaining game for multi-target adversarial training, we first analyze the existing methods, *i.e.*, MAX (Tramer & Boneh, 2019), MSD (Maini et al., 2020), and AVG (Tramer & Boneh, 2019), and categorize them into two types, MAX-based and AVG-based algorithms. We identify a phenomenon where one player dominates the bargaining game at any time $t$, where the agreed gradient at any time $t$ is the same as this player's gradient. Following that, we show that MAX-based algorithms, *i.e.*, MAX and MSD, do not converge. Based on our theoretical results, we propose a novel mechanism which adaptively adjusts the budget of adversaries to avoid any player dominating the bargaining game. We show that with our proposed mechanism, the robust accuracy of MAX, AVG and MSD improves on both the MNIST and CIFAR-10 datasets.

## 1.1 Cooperative Bargaining Game

Cooperative bargaining game (Thomson, 1994) is a process in which several parties jointly decide how to share a surplus that they can jointly gain. The multi-target adversarial training can be viewed as a cooperative game in which each target (perturbation) represents a player, whose utility is derived from its gradient, and all the players negotiate to reach an agreed direction.

In the cooperative bargaining game, we have $K$ players with their own utility function $u_i : \mathcal{A} \bigcup \{\mathbf{d}\} \to \mathbb{R}$, where $\mathcal{A}$ is the set of possible agreements and $\mathbf{d}$ is the disagreement point. The feasible set of utility is defined as $\mathcal{S} = \{(u_1(\boldsymbol{\gamma}) \dots, u_K(\boldsymbol{\gamma})) : \boldsymbol{\gamma} \in \mathcal{A}\}$. The goals of players are to maximize their own utility functions. $\mathcal{S}$ is assumed to be convex and compact throughout this paper while there exists a point $\boldsymbol{\gamma} \in \mathcal{A}$ satisfying $u_i(\boldsymbol{\gamma}) > u_i(\mathbf{d}), \forall i \in [K]$.

We formalize the multi-target adversarial training problem as a bargaining game as follows. This bargaining game has $K$ players, as we are targeting at multi-target adversarial training. For each player, they generate a data-dependent perturbation $\boldsymbol{\delta}_k(\mathbf{x}), \forall k \in [K]$ to complete the adversarial training, where $[K] = \{1, 2, 3, \cdots, K\}$. The possible agreements $\mathcal{A}$ are $\{\sum_{k \in [K]} \gamma_k = 1, \gamma_k \geq 0, \forall k \in [K]\}$ and the disagreement point at $\mathbf{0}$, *i.e.*, staying at the current parameters $w$. The utility function for the player $k$ is $u_k(\boldsymbol{\gamma}) = g_k^\top \sum_{i \in [K]} \gamma_i g_i$, where $g_k$ is the gradient of the player $k$. We note that since the agreement set $\mathcal{A}$ is compact and convex and the utilities are linear, the set of possible payoffs $\mathcal{S}$ is also compact and convex.

## 2 Convergence Analysis

We start the section by showing our theoretical results based on two machine learning models. Furthermore, we design a general framework for the multi-target adversarial training problem to

avoid the player domination phenomenon which might lead to the non-convergence of MAX and MSD in the next section. All missing proofs are in the appendix.

## 2.1 Convergence analysis on SVM model

Consider the binary classification setup introduced in Tsipras et al. (2019), where data $(\mathbf{x}, y)$ is sampled from a distribution $\mathcal{D}$ defined by

$$y \overset{\text{u.a.r}}{\sim} \{+1, -1\}, \quad x_1 = \begin{cases} +y, & \text{w.p. } p; \\ -y, & \text{w.p. } 1-p, \end{cases} \quad x_2, \ldots, x_{d+1} \overset{\text{i.i.d.}}{\sim} \mathcal{N}(\mu y, 1),$$

where $\mathbf{x} = [x_1, \ldots, x_{d+1}] \in \mathbb{R}^{d+1}$, $y$ is a Rademacher random variable, and $\mathcal{N}(\mu, \sigma^2)$ is a normal distribution with mean $\mu$ and variance $\sigma^2$. In our setting, $p \in [0.5, 1]$. $x_1$ is a robust feature while $\mathbf{x}_2, \ldots, \mathbf{x}_{d+1}$ are non-robust features that are weakly correlated with the label. Similarly, we set $\mu$ to be large enough such that almost any classifier can get a high standard accuracy ($> 99\%$), *i.e.*, $\mu \geq 1/\sqrt{d}$. We train a linear model $f_{\mathbf{w}}(\cdot)$ with soft-SVM loss on the data shown above:

$$\min_{\mathbf{w}} \mathbb{E}_{(\mathbf{x}, y) \sim \mathcal{D}} \sum_{p \in \{1, 2, \infty\}} \boldsymbol{\gamma}_p \max\left(0, 1 - y f_{\mathbf{w}}(\mathbf{x} + \boldsymbol{\delta}(\mathbf{x})_p)\right), \text{ s.t. } \|\mathbf{w}\|_2 = 1, \tag{1}$$

where $f_{\mathbf{w}}(\mathbf{x}) = \mathbf{w}^\top \mathbf{x}$, and $\boldsymbol{\gamma} = [\boldsymbol{\gamma}_1, \boldsymbol{\gamma}_2, \boldsymbol{\gamma}_\infty]$ satisfies $\sum_{i \in \{1, 2, \infty\}} \boldsymbol{\gamma}_i = 1$. With the linearity property of SVM, the closed form of optimal perturbations could be calculated as

$$\boldsymbol{\delta}_\infty^*(\mathbf{w}) = -y\epsilon_\infty \operatorname{sign}(\mathbf{w}), \quad \boldsymbol{\delta}_1^*(\mathbf{w}) = \frac{-y\epsilon_1 \mathbf{w}}{\|\mathbf{w}\|_1}, \quad \boldsymbol{\delta}_2^*(\mathbf{w}) = \frac{-y\epsilon_2 \mathbf{w}}{\|\mathbf{w}\|_2}. \tag{2}$$

Let $\mathbf{w}^t$ and $\boldsymbol{\delta}^t$ be the weight vector of classifier and the perturbation at time $t$, respectively. The training procedures of AVG, MAX and MSD are illustrated as follows:

0. Initialize the weights with natural training, *i.e.*, minimizing the soft-SVM loss without perturbation

$$\min_{\mathbf{w}} \mathbb{E}_{(\mathbf{x}, y) \sim \mathcal{D}} \left[\max(0, 1 - y\mathbf{w}^\top \mathbf{x})\right], \quad \text{s.t.} \quad \|\mathbf{w}\|_2 = 1. \tag{3}$$

1. Solve the inner maximization problem using AVG, MAX and MSD, and return the optimal perturbations, *i.e.*, $\boldsymbol{\delta}_1^t, \boldsymbol{\delta}_2^t$ and $\boldsymbol{\delta}_\infty^t$.

2. Update the weight of the classifier by

$$\mathbf{w}^t = \operatorname{argmin}_{\mathbf{w}} \mathbb{E}_{(\mathbf{x}, y) \sim \mathcal{D}} \sum_{p \in \{1, 2, \infty\}} \boldsymbol{\gamma}_p^t \max(0, 1 - y\mathbf{w}^\top(\mathbf{x} + \boldsymbol{\delta}_p^t)), \quad \text{s.t.} \quad \|\mathbf{w}^t\|_2 = 1,$$

where $\boldsymbol{\gamma}^t = [1/3, 1/3, 1/3]$ if the algorithm is AVG; $\boldsymbol{\gamma}^t \in \{[1, 0, 0], [0, 1, 0], [0, 0, 1]\}$ if the algorithm is MAX or MSD.

3. Loop Steps 1 and 2 for predefined epochs or until convergence.

We first present the following negative result:

**Theorem 1.** *Let $\mu \geq 4/\sqrt{d}$, $\epsilon_\infty \geq 2\mu$, $p \leq 0.977$. If one uses MAX and MSD to train the SVM model and $\epsilon_\infty \geq \frac{2}{d}\epsilon_1$ and $\epsilon_\infty \geq \sqrt{\frac{2}{d}}\epsilon_2$, the loss incurred by the $\infty$-player ($\infty$-adversary) is larger than that by the $1$-player ($1$-adversary) and the $2$-player ($2$-adversary) at any time $t$, i.e., $\ell_{\mathbf{w}^t}(\mathbf{x} + \boldsymbol{\delta_\infty}(\mathbf{x})) \geq \max(\ell_{\mathbf{w}^t}(\mathbf{x} + \boldsymbol{\delta_1}(\mathbf{x})), \ell_{\mathbf{w}^t}(\mathbf{x} + \boldsymbol{\delta_2}(\mathbf{x}))), \forall t, \forall \mathbf{x}$.*

This theorem shows that under certain conditions, the SVM learning dynamics will be affected by only one player, *i.e.*, the $\infty$-player, during the whole training procedure. And we further observe that this phenomenon leads to **non-convergence** of the SVM model as the sign of weights of the model will be flipped. We define the phenomenon that one player "dominate" the multi-target adversarial training procedure (the training procedure only depends on one player) as follows

**Definition 2** (Player dominates the cooperative game)**.** *If $\exists i \in [k]$ such that $\boldsymbol{\gamma}_i^t = 1$ and $\boldsymbol{\gamma}_j^t = 0, \forall j \in [K]/\{i\}, \forall t$, then we call that player dominates the bargaining game.*

Further, after analyzing the training dynamics of SVM, we notice that when the $\infty$-player dominates the bargaining game, and given $\epsilon_\infty > \mu$, the SVM model may not converge as the weights for the non-robust features flips over time.

**Theorem 3.** *Consider Problem equation 1 trained with MAX and MSD. If $\infty$-player dominates the bargaining game and $\epsilon_\infty > \mu$, the weights for the non-robust features flips over time,* i.e., $\text{sign}(w_i^t) = -\text{sign}(w_i^{t-1}), \forall i \geq 2$. *Thus, the training procedure does not converge.*

As the $\infty$-player dominates this game, the multi-target adversarial training problem reduces to the single-target problem (equation 6). Further, with Lemma 8, for non-robust feature $i$, if $\eta_\infty > \eta$, we have $\text{sign}(w_i^t) = -\text{sign}(w_i^{t-1})$. Therefore, the training procedure does not converge.

Though we only analyze the case when an $\infty$-player dominates the bargaining game, we notice that when other players dominate this bargaining game (*i.e.*, multi-target adversarial training), with certain conditions, *e.g.*, $\epsilon_1 > 4\mu$, a similar phenomenon can be observed empirically. Motivated by the negative results of the SVM model, we next testify a conjecture that player domination may lead to non-convergence of the linear model as well.

## 2.2 Player Domination Leads to Non-convergence

To testify our conjecture, we introduce a linear model as follows. The linear model is parameterized by $\mathbf{w}$ and optimized by gradient-based algorithms, *e.g.*, AdaGrad (Duchi et al., 2011) or Adam (Kingma & Ba, 2015). The parameter at time (epoch) $t$ is denoted by $\mathbf{w}^t$. The loss function of each player is denoted as $\ell_k, k \in [K]$, which is $L$-smooth and $\mu$-strongly convex, and the corresponding gradient is denoted as $\nabla_{\mathbf{w}} \ell_k(\mathbf{w}^t)$ or $g_k^t, k \in [K], \forall t$. We assume that for a sequence $\{\mathbf{w}^t\}_{t=[1,\infty]}$ generated by any optimization algorithm, the set of the gradient vectors $g_k^t, k \in [K]$ at any time $t$ and at any partial limit are linearly independent unless that point is locally optimal. All loss functions are differentiable and all sub-level sets are bounded. The learning rate is denoted as $\eta$ and $\eta < \frac{2}{L}$. We also assume that the input domain is open and convex.

To generalize our theoretical results, we show that in this linear model, MAX and MSD still may not converge if one player dominates the game.

**Theorem 4.** *Consider using MAX and MSD to train the linear model described above. If the bargaining game is dominated by one player during the whole game (see Definition 2), then the loss of all players and the overall loss would increase as time $t$ grows. That means the training procedure on the above linear model might not converge.*

While we have shown that MAX and MSD may not converge under the two models that we study, we notice that AVG provably converges as the loss is decreasing w.r.t the number of epochs. See the following theorem.

**Theorem 5.** *Using AVG to train the linear model, the overall loss decrease as time $t$ grows.*

This theorem shows that under the same setting, while the loss of each player and the overall loss will be increasing as time grows with MSD and MSD, the overall loss will be decreasing with AVG. And the key component in the proof is the player domination phenomenon, which directly leads to the increase of loss with MAX and MSD. That indicates that the training procedure with MAX and MSD might not converge while AVG might converge, which is also a consequence of player domination.

## 3 Avoiding Player Domination with Adaptive Budgets

Our theoretical results indicate that when player domination (Def. 2) occurs, MAX and MSD may not converge while the loss of AVG decreases as time grows. Inspired by this analysis, we design a novel general-purpose algorithm for multi-target adversarial robustness, which adaptively changes the budget of different adversaries. The resulting **AdaptiveBudget** method is presented in Algorithm 1.

The core idea of this algorithm is to avoid player domination by adaptively assigning proper attack budgets to different adversaries (players). Such an assignment is to make no single player achieve significantly better than others. Concretely, for each batch of data, we first get the adversarial perturbations $\boldsymbol{\delta}_\infty, \boldsymbol{\delta}_1$ and $\boldsymbol{\delta}_2$ for the $\ell_\infty, \ell_1$, and $\ell_2$ adversaries. Then based on the norms of the gradients by forwarding their adversarial examples through our model, the algorithm adaptively adjusts the budgets $\epsilon$ for different adversaries to avoid the player domination phenomenon. Specifically, our

---

**Algorithm 1** Framework of Multi-target Adversarial Training with Adaptive Budget

---

**Require:** Training Epochs $E$, Training samples $(\mathcal{X}, \mathcal{Y})$, adversarial budgets $(\epsilon_\infty, \epsilon_1, \epsilon_2)$, model $f(\cdot)$, loss function $\ell$.

1: **for** $e \in [E]$ **do**
2:      **for** $\mathbf{x}, y \in (\mathcal{X}, \mathcal{Y})$ **do**
3:          $g_p \leftarrow \ell'(\mathbf{x} + \boldsymbol{\delta}_p(\mathbf{x})), \boldsymbol{\delta}_p(\mathbf{x}) \leftarrow \mathrm{PGD}(\mathbf{x}, k, \eta, \ell, \epsilon_p, \ell), \forall p \in \{1, 2, \infty\}$
4:          Get adaptive budgets $\hat{\epsilon}_1, \hat{\epsilon}_2, \hat{\epsilon}_\infty \leftarrow$ **AdaptiveBudget**$([g_1, g_2, g_\infty], [\epsilon_1, \epsilon_2, \epsilon_\infty])$;
5:          Adversarial training using MAX, MSD or AVG with budgets $(\hat{\epsilon}_1, \hat{\epsilon}_2, \hat{\epsilon}_\infty)$;
6:      **end for**
7: **end for**
8: **Return** the classifier $f$.
9:
10: **AdaptiveBudget**(Gradients$[g_1, g_2, g_\infty]$, Epsilon$[\epsilon_1, \epsilon_2, \epsilon_\infty]$):
11:      $p_{\max} \leftarrow \arg\max_{p \in \{\infty, 1, 2\}} \|g_p\|, \quad p_{\min} \leftarrow \arg\min_{p \in \{\infty, 1, 2\}} \|g_p\|$;
12:      $p_{\mathrm{mid}} \leftarrow \{1, 2, \infty\} / \{p_{\max}, p_{\min}\}$;
13:      $\epsilon_{p_{\max}} \leftarrow \epsilon_{p_{\max}} \cdot \frac{\|g_{p_{\max}}\|}{\|g_{p_{\mathrm{mid}}}\|}, \quad \epsilon_{p_{\min}} \leftarrow \epsilon_{p_{\min}} \cdot \frac{\|g_{p_{\min}}\|}{\|g_{p_{\mathrm{mid}}}\|}$;
14:      **Return** $\epsilon_1, \epsilon_2, \epsilon_\infty$.

---

proposed method keeps the budget of the adversary whose norm of gradient is the middle one. This increases the budget of the adversary whose norm of gradient is the maximum and decreases the budget of the adversary whose norm of gradient is the minimum. The intuition behind our method is to focus on the hardest task in the current round so that this task might be easier to model in the next round. After obtaining the adjusted adversarial budgets, the model utilizes MSD, MAX or AVG to approximately solve the inner maximization problem and then updates its parameter with any gradient descent algorithm.

The proposed framework is general and can be applied to all existing multi-target adversarial training algorithms. The adaptive budget module is employed to break the curse of player domination which might occur when applying MAX and MSD to train a robust model. In the next section, we corroborate the consistent effectiveness of **AdaptiveBudget** method with extensive experiments.

## 4 Experiments

### 4.1 Experimental setup and implementation details

**Datasets.** We conducted extensive experiments on a synthetic data (Sec. 2.1) to complement our theoretical results and MNIST (LeCun et al., 1998) and CIFAR-10 (Krizhevsky & Hinton, 2009) to show the superiority of our proposed methods over the existing methods of multi-target adversarial training. **Methods.** We train models that defend multiple adversaries using MAX (Tramer & Boneh, 2019), AVG (Tramer & Boneh, 2019) and MSD (Maini et al., 2020). **Attacks used for evaluation.** To fully understand the performance of the defense, we employ PGD adversary and Autoattack (Croce & Hein, 2020b)[1] to test the effectiveness of our method. We make 10 random restarts for presenting all the results presented later for both MNIST and CIFAR-10. The budgets for three adversaries, *i.e.*, $\epsilon_1$, $\epsilon_2$, and $\epsilon_\infty$, are the same with the setting at training for both two datasets, while for the number of iterations, we increase them to $(100, 200, 100)$ for $(\ell_\infty, \ell_2, \ell_1)$ on MNIST and $(100, 500, 100)$ for $(\ell_\infty, \ell_2, \ell_1)$ on CIFAR-10.

### 4.2 Results on MNIST

Here we present results on the MNIST dataset, summarized in Table 1. Though MNIST has been treated as an "easy" benchmark compared with CIFAR-10 or bigger datasets, *e.g.*, ImageNet (Deng et al., 2009), we notice that all the single target adversarial training methods, *i.e.*, $\ell_1$, $\ell_2$, and $\ell_\infty$, fail to defend only three attacks, while the best is $\ell_\infty$ training which defends almost three attacks and outperforms the MAX on the overall Accuracy. From Table 1, with our proposed method, we notice that the robust accuracy against the $\ell_\infty$ PGD attack is improved with three methods, *i.e.*, MAX, MSD, and AVG, using both $\ell_1$ and $\ell_2$ norms. The $\ell_1$ and $\ell_2$ robust accuracy against $\ell_1$ and $\ell_2$ PGD attacks is improved on MAX by $4.3\%$ and $2.2\%$ (with adaptive budget using $\ell_1$ norm), $2.6\%$ and $2.3\%$ (with adaptive budget using $\ell_2$ norm) as well. Maini et al. (2020) and Tramer & Boneh (2019) mentioned

---

[1]We only consider white-box attack based on gradient, while not using the attack based on gradient estimation, as the gradient for the standard architectures used here are readily available.

Table 1: Summary of robust accuracy for MNIST (higher is better). "w. adaptive budget" refers to employing our proposed framework which enables adaptive budget to avoid any player dominating the game. "*" means that the results are reproduced from the implementation of Maini et al. (2020) by the hyperparameters of Maini et al. (2020). "$\ell_1$ (ours)" and "$\ell_2$ (ours)" refers to employing our proposed adaptive budget method w.r.t $\ell_1$ and $\ell_2$ norms.

| Models w. adaptive budget | $\ell_1$ | $\ell_2$ | $\ell_\infty$ | MAX | $\ell_1$ (ours) | $\ell_2$ (ours) | MSD | $\ell_1$ (ours) | $\ell_2$ (ours) | AVG | $\ell_1$ (ours) | $\ell_2$ (ours) |
|---|---|---|---|---|---|---|---|---|---|---|---|---|
| Clean Accuracy (%) | 97.2 | 99.1 | 99.2 | 98.6 | 98.9 | 98.9 | 98.2 | 98.3 | 98.9 | 99.1 | 99.1 | 99.1 |
| $\ell_1$ PGD Robust Acc (%) | 47.3* | 67.8* | 54.6* | 67.1* | **71.4**↑ | **69.7**↑ | 67.3* | 66.8↓ | 65.9↓ | 70.6* | 68.2↓ | 68.9↓ |
| $\ell_2$ PGD Robust Acc (%) | 24.1* | 66.8* | 61.8* | 67.2* | **69.4**↑ | **69.5**↑ | 68.0* | 67.9↓ | 65.3↓ | 69.4* | 68.3↓ | 68.3↓ |
| $\ell_\infty$ PGD Robust Acc (%) | 0* | 0.1* | 88.9* | 21.2* | **67.2**↑ | **67.6**↑ | 62.4* | **69.7**↑ | **69.7**↑ | 59.5* | **67.7**↑ | 65.6↑ |
| All PGD Robust Acc (%) | 0* | 0.1* | 52.1* | 21.2* | **61.3**↑ | **61.4**↑ | 59.7* | **62.1**↑ | **61.0**↑ | 55.4* | **59.2**↑ | 58.2↑ |

Table 2: Summary of robust accuracy for CIFAR-10 (higher is better). "w. adaptive budget" refers to employing our proposed framework which enables adaptive budget to avoid any player dominating the game. "AA" refers to AutoAttack. "*" means that the results are reproduced from the implementation of Maini et al. (2020) by the hyperparameters of Maini et al. (2020). "$\ell_1$ (ours)" and "$\ell_2$ (ours)" refers to employing our proposed adaptive budget method w.r.t $\ell_1$ and $\ell_2$ norms.

| Models w. adaptive budget | $\ell_1$ | $\ell_2$ | $\ell_\infty$ | MAX | $\ell_1$ (ours) | $\ell_2$ (ours) | MSD | $\ell_1$ (ours) | $\ell_2$ (ours) | AVG | $\ell_1$ (ours) | $\ell_2$ (ours) |
|---|---|---|---|---|---|---|---|---|---|---|---|---|
| Clean Accuracy | 92.4 | 87.5 | 84.2 | 79.6 | 76.9 | 78.7 | 79.2 | 77.6 | 79.0 | 83.8 | 81.6 | 81.5 |
| $\ell_1$ PGD Robust Acc (%) | 90.8 | 31.7 | 17.3 | 44.0* | **50.7**↑ | **51.7**↑ | 50.8* | **51.2**↑ | **52.6**↑ | 55.7* | **57.3**↑ | **56.3**↑ |
| $\ell_2$ PGD Robust Acc (%) | 0.1 | 64.0 | 60.6 | 55.6* | **63.4**↑ | **65.1**↑ | 64.3* | 63.6↓ | **65.5**↑ | 67.0* | 66.6↓ | 67.0 |
| $\ell_\infty$ PGD Robust Acc (%) | 0 | 27.8 | 51.2 | 41.3* | **47.5**↑ | **47.6**↑ | 45.7* | **48.4**↑ | **47.2**↑ | 39.4* | **45.5**↑ | **44.2**↑ |
| All PGD Robust Acc (%) | 0 | 23.8 | 17.3 | 40.4* | **46.0**↑ | **46.8**↑ | 44.1* | **47.2**↑ | **46.4**↑ | 39.2* | **45.2**↑ | **43.6**↑ |
| $\ell_1$ AA Robust Acc (%) | 0 | 23.8 | 6.2 | 41.4* | **45.7**↑ | **45.5**↑ | 45.5* | **46.4**↑ | **46.7**↑ | 49.7* | **52.7**↑ | **50.8**↑ |
| $\ell_2$ AA Robust Acc (%) | 0 | 63.0 | 57.4 | 53.7* | **60.4**↑ | **63.2**↑ | 61.9* | **62.3**↑ | **62.1**↑ | 65.4* | 64.6↓ | 65.5↑ |
| $\ell_\infty$ AA Robust Acc (%) | 0 | 26.1 | 48.0 | 38.4* | **44.7**↑ | **44.1**↑ | 43.1* | **45.2**↑ | **44.4**↑ | 37.0* | **43.1**↑ | **42.1**↑ |
| All AA Robust Acc (%) | 0 | 19.5 | 6.2 | 37.6* | **42.9**↑ | **42.3**↑ | 41.6* | **43.4**↑ | **43.0**↑ | 36.6* | **42.5**↑ | **41.2**↑ |

that there is a trade-off between the robust accuracy against the $\ell_\infty$ attack and the robust accuracy against the $\ell_1$ and $\ell_2$ attacks. Similar observations can be obtained from our experimental results. For MSD and AVG, the robust accuracy defending $\ell_1$ and $\ell_2$ PGD attacks drops a bit, which might be due to that trade-off. We also observe that the proposed adaptive budget method avoids the player domination phenomenon well as it improves the robust accuracy of MAX by approximately $40\%$. The all PGD robust accuracy of the vanilla MAX also shows that the player domination phenomenon hinders MAX from achieving a satisfying robust accuracy for the non-convex scenarios as well.

### 4.3 Results on CIFAR-10

The results are shown in Table 2, and the curve of robust accuracy is shown in Figures 5 and 4. Due to the limitation of space, we present the most important results in the main paper while leaving the left results in the Appendix.

**Main Results.** The results presented in Table 2 show the generalization ability of our proposed method, which improves the robust accuracy of three methods, *i.e.*, MSD, MAX, and AVG, against $\ell_1$, $\ell_2$, $\ell_\infty$, and all attacks by PGD and AutoAttack. We notice that the robust accuracy of all attacks for PGD and AutoAttack is mainly restricted by how well the model defends the $\ell_\infty$ attack. It might be caused by the fact that the radius of $\ell_\infty$ attack might be too small compared with the radius of $\ell_1$ and $\ell_2$ attacks such that with the updates by gradient-based algorithms, the gradient of $\ell_\infty$ adversary is covered by others. In this way, the model ignores the $\ell_\infty$ adversary. And we notice that employing the adaptive budget with either the $\ell_1$ or $\ell_2$ norm helps models to pay attention to the tasks that are not well learnt as the robust accuracy on $\ell_\infty$ adversary is relatively improved by the most. For example, the $\ell_\infty$ PGD robust accuracy of MAX with the adaptive budget w.r.t the $\ell_1$ norm experiences a relative $15.01\%$ improvement while there is only a $14.03\%$ relative improvement on the $\ell_2$ PGD robust accuracy. Besides, the trade-off between the three attacks is different from the trade-off on MNIST. On MNIST, the robust accuracy of $\ell_2$ adversary is related to that of $\ell_1$ adversary while on CIFAR-10, it seems that $\ell_2$ robust accuracy is more likely to relate to $\ell_\infty$ robust accuracy.

# 5  Conclusion

In this paper, to achieve the ultimate goal of robustness, *i.e.*, defending any terms of attacks, we first formalized this problem within the scope of the bargaining game, and investigated the convergence property of MAX, MSD, and AVG under two linear cases. We found that MSD and MSD did not converge theoretically due to a phenomenon named player domination while AVG did not suffer from this. To avoid player domination in the training of robust models, we designed a novel framework for multi-target adversary training with an adaptive budget method. Specifically, the adaptive budget method adaptively changed the budget of different attacks to avoid player domination. Finally, to evaluate the proposed framework, we conducted experiments on two benchmarks, *i.e.*, MNIST and CIFAR-10. Experimental results showed that our proposed adaptive budget method improved the robust accuracy on two benchmarks, which complemented our theoretical results and also supported our finding that player domination might interfere with the training of robust models.

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

# A Related work

**Adversarial Training.** Goodfellow et al. (2015) has shown that a single, small step in the direction of the sign of the gradient may fool deep learning models for the image classification task. Later it was extended to a multi-step attack (Kurakin et al., 2017) namely the Basic Iterative Method (now typically referred to as the PGD attack), which significantly improves the success of creating adversarial examples. After that, different variations of PGD attack (Brendel et al., 2019; Li et al., 2019; Croce & Hein, 2020b) have been proposed to overcome heuristic defenses and create stronger adversaries. To defend the adversaries, numerous defend methods (Papernot et al., 2016; Kannan et al., 2018; Madry et al., 2018; Zhang et al., 2019b,a, 2020; Wu et al., 2020; Wang et al., 2020; Pang et al., 2020b; Shi et al., 2020; Zhang et al., 2022a) have been discovered. Among these methods, the most successful defense method is adversarial training (Madry et al., 2018), which formulates the defense problem as a min-max optimization problem and has become one of the few adversarial defenses that is still robust to the following stronger attacks (Carlini & Wagner, 2017; Athalye et al., 2018; Mosbach et al., 2018). Following that, the empirical robustness (Pang et al., 2020a, 2021; Gao et al., 2022; Zhang et al., 2022b; Wang et al., 2021) has been significantly advanced during the past few decades.

**Multi-target Adversarial Training.** Robustness against multiple types of attacks simultaneously relates to our work very closely. In 2019, Schott et al. (2019) used multiple variational autoencoders and constructed an architecture named analysis by synthesis for the MNIST dataset. Their experimental results showing that even for MNIST, it is hard to train a model robust to three different adversaries. Following that, (Tramer & Boneh, 2019) investigated the theoretical and empirical trade-offs of adversarial robustness when defending against aggregations of multiple adversaries. Their results showed that the model robust to the $\ell_\infty$ adversary might not be able to defend other attacks, *i.e.*, $\ell_1$ and $\ell_2$ attacks, on MNIST. To alleviate this problem, they designed a augmentation-based method to get $\ell_2$ robustness instead of using the $\ell_2$ ball. Later, Croce & Hein (2020a) proposed a provable adversarial defense against all $\ell_p$ norms for $p \geq 1$ with regularization methods. From a greedy search perspective, Maini et al. (2020) suggested that taking the worst-case over all steepest descent directions helps achieve better performance than MAX and AVG empirically. Recently, while not studied as a defense method, (Kang et al., 2019a) investigated the transferability of adversarial robustness between models trained against different perturbation models.

# B Preliminaries

## B.1 Problem Formulation

The goal of multi-target adversarial training is to learn a function $f_\mathbf{w} : \mathcal{X} \rightarrow \{-1, +1\}$ that is robust to adversarial examples generated by multiple adversaries[2], where $f$ is parameterized by $\mathbf{w}$. The multi-target robust loss of $f_\mathbf{w}$ is defined as $\mathbb{E}_{(\mathbf{x},y)}[\max_{\boldsymbol{\delta}\in\mathcal{B}} \ell(f_\mathbf{w}(\mathbf{x} + \boldsymbol{\delta}), y)]$ where $\mathcal{B} = \mathcal{B}_1(\epsilon_1) \bigcup \mathcal{B}_2(\epsilon_2) \bigcup \mathcal{B}_\infty(\epsilon_\infty)$ and $\mathcal{B}_p(\epsilon) = \{\boldsymbol{\delta} : \|\boldsymbol{\delta}\|_p \leq \epsilon\}$, and $\boldsymbol{\delta}$ is the perturbation. In deep learning scenarios, Adversarial Training (Madry et al., 2018) is frequently used to train a robust classifier. Previous multi-target adversarial training work, *e.g.*, MSD, MAX and AVG, employs the following minimax objective to update the algorithm,

$$\min_\mathbf{w} \mathbb{E}_{(\mathbf{x},y)} \max_{\boldsymbol{\delta}\in\mathcal{B}} \ell(f_\mathbf{x}(\mathbf{x} + \boldsymbol{\delta}), y) \,. \tag{4}$$

In the deep learning scenario, as this problem is often non-convex and hard to derive the closed-form solution, the above objective is optimized by iteratively optimizing $\boldsymbol{\delta}$ and $\mathbf{w}$ for several rounds as shown below,

1. Algorithm gets the optimal perturbation $\boldsymbol{\delta}^t$ for the $t$-th round as

$$\boldsymbol{\delta}^t(\mathbf{x}) = \mathrm{argmax}_{\boldsymbol{\delta}\in\mathcal{B}} \ell(f_{\mathbf{w}^{t-1}}(\mathbf{x} + \boldsymbol{\delta}), y) \,. \tag{5}$$

2. Algorithm gets the optimal parameter $w^t$ for the $t$-th round as

$$\mathbf{w}^t = \mathrm{argmin}_\mathbf{w} \mathbb{E}_{(\mathbf{x},y)} \ell(f_\mathbf{x}(\mathbf{x} + \boldsymbol{\delta}^t(\mathbf{x})), y) \,.$$

---

[2]In our paper, we analyze the case where three adversaries are involved, *i.e.*, $\ell_1$, $\ell_2$ and $\ell_\infty$.

---

**Algorithm 2** MAX, AVG and MSD algorithms

---

1: **MAX**(input data $\mathbf{x}$, steps $k$, stepsize $\eta$, perturbation budgets $(\epsilon_\infty, \epsilon_1, \epsilon_2)$, loss function $\ell$):
2:     $\boldsymbol{\delta}_p(\mathbf{x}) = \mathrm{PGD}(\mathbf{x}, k, \eta, \ell_p, \epsilon_p, \ell), p \in \{1, 2, \infty\}$;
3:     **Return** $\mathrm{argmax}_{\boldsymbol{\delta} \in \{\boldsymbol{\delta}_1(x), \boldsymbol{\delta}_2(x), \boldsymbol{\delta}_\infty(x)\}} \ell(\mathbf{x} + \boldsymbol{\delta}, y)$.

4:
5: **AVG**(input data $\mathbf{x}$, steps $k$, stepsize $\eta$, perturbation budgets $(\epsilon_\infty, \epsilon_1, \epsilon_2)$, loss function $\ell$):
6:     **Return** $\{\boldsymbol{\delta}_p(\mathbf{x}) = \mathrm{PGD}(\mathbf{x}, k, \eta, \ell_p, \epsilon_p, \ell)\}_{p \in \{1, 2, \infty\}}$.

7:
8: **MSD**(input data $\mathbf{x}$, steps $k$, stepsize $\eta$, perturbation budgets $(\epsilon_\infty, \epsilon_1, \epsilon_2)$, loss function $\ell$):
9:     $\boldsymbol{\delta}^0 = 0$;
10:     **for** $i \in [k]$ **do**
11:         $\boldsymbol{\delta}_p^i = \mathrm{PGD}_{\mathrm{Step}}(\mathbf{x}, \boldsymbol{\delta}^i, \eta, \ell_p, \epsilon_p, \ell)$;
12:         $\boldsymbol{\delta}^{i+1} = \mathrm{argmax}_{\boldsymbol{\delta}_p \in \{\boldsymbol{\delta}_1^i, \boldsymbol{\delta}_2^i, \boldsymbol{\delta}_\infty^i\}} loss(\mathbf{x} + \boldsymbol{\delta}_p^t, y)$;
13:     **end for**
14:     **Return** $\boldsymbol{\delta}^k$.

---

To find the approximate optimal perturbation $\boldsymbol{\delta}$ and the approximate optimal parameter $w$ under the non-convex scenario, stochastic gradient descent and projected gradient descent (PGD) methods are used to approximately solve the above minimization on $w$ and maximization on $\boldsymbol{\delta}$ problems. Specifically, PGD runs several predefined PGD step $\mathbf{PGD}_{\mathrm{step}} = \mathrm{Proj}_{\mathrm{Ball}_p(0, \epsilon_p)}(\boldsymbol{\delta} + \eta \, \mathrm{sign}(\nabla \ell(\mathbf{x} + \boldsymbol{\delta}, y)))$ to approximately find a worst-case adversarial example, where $\nabla \ell(\mathbf{x} + \boldsymbol{\delta}, y)$ is the gradient of $\ell(\mathbf{x} + \boldsymbol{\delta}, y)$.

Tramer & Boneh (2019) first proposed to solve the inner maximization problem of the problem equation 4, *i.e.*, problem (5), by the MAX (the worst-case perturbation, Algorithm 2) and AVG (augmentation of all perturbations, Algorithm 2) as below

$$\text{MAX: } \mathbb{E}_{(\mathbf{x}, y)} \ell(f_{\mathbf{w}}(\mathbf{x} + \mathrm{MAX}(\mathbf{x})), y), \quad \text{AVG: } \mathbb{E}_{(\mathbf{x}, y)} \frac{1}{3} \sum_{\boldsymbol{\delta} \in AVG(\mathbf{x})} \ell(f_{\mathbf{w}}(\mathbf{x} + \boldsymbol{\delta}), y),$$

where $\boldsymbol{\delta}_1, \boldsymbol{\delta}_2$ and $\boldsymbol{\delta}_\infty$ are obtained by $l_1, l_2$ and $l_\infty$ PGD adversaries.

Later, Maini et al. (2020) designed a "greedy" algorithm named MSD, which solves the inner maximization problem by simultaneously maximizing the worst-case loss overall perturbation models at each projected steepest descent step as shown in Algorithm 2. And then the inner maximization becomes as follows

$$\mathbb{E}_{(\mathbf{x}, y)} \ell(f_{\mathbf{w}}(\mathbf{x} + \mathrm{MSD}(\mathbf{x})), y).$$

## C Additional Lemmas

We denote the following single-target adversarial training problem

$$\min_w \mathbb{E}_{(x, y) \sim \mathcal{D}} \max(0, 1 - yw^\top(x + \boldsymbol{\delta}_p^*)) \tag{6}$$
$$\text{s.t.} \|w\|_2 = 1,$$

where $p \in \{1, 2, \infty\}$ is given before the training procedure.

**Lemma 6** (lemma D.1 (Tsipras et al., 2019)). *The optimal solution* $\mathbf{w}^* = (w_1, ..., w_{d+1})$ *of our optimization problem equation 3 must satisfy* $w_2 = ... = w_{d+1}$ *and* $sign(w_2) = sign(\mu)$.

**Lemma 7** (lemma D.2 (Tsipras et al., 2019)). *The optimal solution* $\mathbf{w}^* = (w_1, ..., w_{d+1})$ *of our optimization problem equation 3 must satisfy* $w_1 \leq 1/\sqrt{2}$ *and* $w_2 = ... = w_{d+1} \geq 1/\sqrt{2d}$.

**Lemma 8.** *In the adversarial training framework, for arbitrary step $t$, if $\epsilon > \mu$ and*

$$p \leq 1 - \max(\frac{\mathbb{E}[\max(0, 1 - \mathcal{N}((\epsilon - \mu)\sqrt{d}, 1))]}{\mathbb{E}[\max(0, 1 + 1/\sqrt{2}(1 + \epsilon) - \mathcal{N}((\epsilon - \mu)\sqrt{\frac{d}{2}}, 0.5))]},$$

$$\frac{\mathbb{E}[\max(0, 1 - \mathcal{N}((\epsilon + \mu)\sqrt{d}, 1))]}{\mathbb{E}[\max(0, 1 + 1/\sqrt{2}(1 + \epsilon) - \mathcal{N}((\epsilon + \mu)\sqrt{\frac{d}{2}}, 0.5))]}),$$

*the optimal solution* $\mathbf{w}_t^* = (w_t^1, ..., w_t^{d+1})$ *of our optimization problem must satisfy* $w_t^1 \leq 1/\sqrt{2}$ *and* $w_t^2 = ... = w_t^{d+1}$ *and* $|w_t^2| \geq 1/\sqrt{2d}$. *Moreover,* $sign(w_t^i) = -sign(w_{t+1}^i), i \in [2, d+1]$

*Proof.* $t = 0$, by Theorem 7 the result holds and $sign(w_0^i) = sign(\mu), i \in [2, d+1]$.

$t = 1$, the perturbed distribution is given by

$$y \sim \{-1, 1\}, \quad x_1 \sim \begin{cases} y(1 - \epsilon), & \text{with prob } p; \\ -y(1 + \epsilon), & \text{with prob } 1 - p, \end{cases} \quad x_i \sim \mathcal{N}((\mu - \epsilon)y, 1), \; i \geq 2$$

Assume for the sake of contradiction that $w_1^1 \geq 1/\sqrt{2}$, by Theorem 6 we have $0 \geq w_1^2 = ... = w_1^{d+1} \geq -1/\sqrt{2d}$. Then, with probability at least $1 - p$, the first feature predicts the wrong label and without enough weight, the remaining features cannot compensate for it. Concretely,

$$\mathbb{E}[\max(0, 1 - y\mathbf{w}_1^{*T}(\mathbf{x} - \boldsymbol{\delta}_\infty))] \geq (1 - p)\mathbb{E}[\max(0, 1 + w_1^1(1 + \epsilon) - |w_1^2|\sum_{i=2}^{d+1}\mathcal{N}(\epsilon - \mu, 1))]$$

$$\geq (1 - p)\mathbb{E}[\max(0, 1 + 1/\sqrt{2}(1 + \epsilon) - \mathcal{N}((\epsilon - \mu)\sqrt{\frac{d}{2}}, 0.5))]$$

We will now show that a solution that assigns zero weight on the first feature ($w_1^2 = 1/\sqrt{d}$ and $w_1^1 = 0$), achieves a better margin loss,

$$\mathbb{E}[\max(0, 1 - y\mathbf{w}_1(\mathbf{x} - \boldsymbol{\delta}_\infty))] = \mathbb{E}[\max(0, 1 - \mathcal{N}((\epsilon - \mu)\sqrt{d}, 1))]$$

Because

$$p \leq 1 - \frac{\mathbb{E}[\max(0, 1 - \mathcal{N}((\epsilon - \mu)\sqrt{d}, 1))]}{\mathbb{E}[\max(0, 1 + 1/\sqrt{2}(1 + \epsilon) - \mathcal{N}((\epsilon - \mu)\sqrt{\frac{d}{2}}, 0.5))]},$$

we have $\mathbb{E}[\max(0, 1 - y\mathbf{w}_1^{*T}(\mathbf{x} - \boldsymbol{\delta}_\infty))] \geq \mathbb{E}[\max(0, 1 - y\mathbf{w}_1(\mathbf{x} - \boldsymbol{\delta}_\infty))]$, which yields contradiction. Besides, in this case $sign(w_1^i) = sign(\mu - \epsilon) = -sign(\mu) = -sign(w_0^i), i \in [2, d+1]$

$t = 2$, the perturbed distribution is given by

$$y \sim \{-1, 1\}, \quad x_1 \sim \begin{cases} y(1 - \epsilon), & \text{with prob } p; \\ -y(1 + \epsilon), & \text{with prob } 1 - p, \end{cases} \quad x_i \sim \mathcal{N}((\mu + \epsilon)y, 1), \; i \geq 2$$

Assume for the sake of contradiction that $w_2^1 \geq 1/\sqrt{2}$, by Theorem 6 we have $0 \geq w_2^2 = ... = w_2^{d+1} \geq -1/\sqrt{2d}$. Then, with probability at least $1 - p$, the first feature predicts the wrong label and without enough weight, the remaining features cannot compensate for it. Concretely,

$$\mathbb{E}[\max(0, 1 - y\mathbf{w}_2^{*T}(\mathbf{x} - \boldsymbol{\delta}_\infty))] \geq (1 - p)\mathbb{E}[\max(0, 1 + w_2^1(1 + \epsilon) - |w_2^2|\sum_{i=2}^{d+1}\mathcal{N}(\epsilon + \mu, 1))]$$

$$\geq (1 - p)\mathbb{E}[\max(0, 1 + 1/\sqrt{2}(1 + \epsilon) - \mathcal{N}((\epsilon + \mu)\sqrt{\frac{d}{2}}, 0.5))]$$

We will now show that a solution that assigns zero weight on the first feature ($w_2^2 = 1/\sqrt{d}$ and $w_2^1 = 0$), achieves a better margin loss.

$$\mathbb{E}[\max(0, 1 - y\mathbf{w}_2(\mathbf{x} - \boldsymbol{\delta}_\infty))] = \mathbb{E}[\max(0, 1 - \mathcal{N}((\epsilon + \mu)\sqrt{d}, 1))]$$

Because

$$p \leq 1 - \frac{\mathbb{E}[\max(0, 1 - \mathcal{N}((\epsilon + \mu)\sqrt{d}, 1))]}{\mathbb{E}[\max(0, 1 + 1/\sqrt{2}(1 + \epsilon) - \mathcal{N}((\epsilon + \mu)\sqrt{\frac{d}{2}}, 0.5))]},$$

we have $\mathbb{E}[\max(0, 1 - y\mathbf{w}_1^{*T}(\mathbf{x} - \boldsymbol{\delta}_\infty))] \geq \mathbb{E}[\max(0, 1 - y\mathbf{w}_1(\mathbf{x} - \boldsymbol{\delta}_\infty))]$, which yields contradiction. Besides, in this case $sign(w_2^i) = sign(\mu + \epsilon) = sign(\mu) = -sign(w_1^i), i \in [2, d+1]$

By induction we can easily derive that $w_t^1 \leq 1/\sqrt{2}$, $w_t^2 = ... = w_t^{d+1}$, $|w_t^2| \geq 1/\sqrt{2d}$ and $sign(w_t^i) = -sign(w_{t+1}^i), i \in [2, d+1]$ for all $t \geq 0$. $\qquad\square$

**Lemma 9.** *If $z \sim \mathcal{N}(\mu, \sigma^2)$,*

$$\mathbb{E}_z[z\mathbb{I}_{z\geq 0}] = \int_0^\infty z\frac{1}{\sqrt{2\pi\sigma^2}}\exp(-\frac{(z-\mu)^2}{2\sigma^2})dz = \frac{\sigma}{\sqrt{2\pi}}\exp(-\frac{\mu^2}{2\sigma^2}) + \frac{\mu}{2}(\text{erf}(\frac{\mu}{\sqrt{2}\sigma}) + 1)$$

**Lemma 10.** *When $\mu \geq 4/\sqrt{d}$, $\epsilon \geq 2\mu$, and $p \leq 0.977$, the optimal solution $\mathbf{w}_t^* = (w_t^1, ..., w_t^{d+1})$ of our optimization problem must satisfy $w_t^1 \leq 1/\sqrt{2}$ and $w_t^2 = ... = w_t^{d+1}$ and $|w_t^2| \geq 1/\sqrt{2d}$.*

*Proof.* Let $\mu = m/\sqrt{d}, m \geq 4, \epsilon = k\mu, k \geq 2$

$$\frac{\mathbb{E}[\max(0, 1 - \mathcal{N}((\epsilon - \mu)\sqrt{d}, 1))]}{\mathbb{E}[\max(0, 1 + 1/\sqrt{2}(1 + \epsilon) - \mathcal{N}((\epsilon - \mu)\sqrt{\frac{d}{2}}, 0.5))]}$$

$$= \frac{\mathbb{E}[\max(0, \mathcal{N}(1 + m - km, 1))]}{\mathbb{E}[\max(0, \mathcal{N}(1 + (1 + m)/\sqrt{2} + km/\sqrt{2d} - km/\sqrt{2}, 0.5))]}$$

$$\leq \frac{\mathbb{E}[\max(0, \mathcal{N}(1 + m - km, 1))]}{\mathbb{E}[\max(0, \mathcal{N}(1 + (1 + m - km)/\sqrt{2}, 0.5))]}$$

$$\frac{\mathbb{E}[\max(0, 1 - \mathcal{N}((\epsilon + \mu)\sqrt{d}, 1))]}{\mathbb{E}[\max(0, 1 + 1/\sqrt{2}(1 + \epsilon) - \mathcal{N}((\epsilon + \mu)\sqrt{\frac{d}{2}}, 0.5))]}$$

$$= \frac{\mathbb{E}[\max(0, \mathcal{N}(1 - m - km, 1))]}{\mathbb{E}[\max(0, \mathcal{N}(1 + (1 - m)/\sqrt{2} + km/\sqrt{2d} - km/\sqrt{2}, 0.5))]}$$

$$\leq \frac{\mathbb{E}[\max(0, \mathcal{N}(1 - m - km, 1))]}{\mathbb{E}[\max(0, \mathcal{N}(1 + (1 - m - km)/\sqrt{2}, 0.5))]}$$

Consider the function $h(a) = \frac{\mathbb{E}[\max(0, \mathcal{N}(a, 1))]}{\mathbb{E}[\max(0, \mathcal{N}(1 + a/\sqrt{2}, 0.5))]} = \frac{\frac{1}{\sqrt{2\pi}}\exp(-\frac{a^2}{2}) + \frac{a}{2}(\text{erf}(\frac{a}{\sqrt{2}}) + 1)}{\frac{1}{2\sqrt{\pi}}\exp(-(1 + \frac{a}{\sqrt{2}})^2) + \frac{1 + \frac{a}{\sqrt{2}}}{2}(\text{erf}((1 + \frac{a}{\sqrt{2}})) + 1)}$

$$h'(a) = ((\frac{1}{2} + \frac{1}{2}\text{erf}(\frac{a}{\sqrt{2}}))(\frac{1}{2\sqrt{\pi}}\exp(-(1 + \frac{a}{\sqrt{2}})^2) + \frac{1 + \frac{a}{\sqrt{2}}}{2}(\text{erf}((1 + \frac{a}{\sqrt{2}})) + 1)) - (\frac{1}{2\sqrt{2}}$$

$$+ \frac{1}{2\sqrt{2}}\text{erf}(1 + \frac{a}{\sqrt{2}}))(\frac{1}{\sqrt{2\pi}}\exp(-\frac{a^2}{2}) + \frac{a}{2}(\text{erf}(\frac{a}{\sqrt{2}}) + 1))))/(\frac{1}{2\sqrt{\pi}}\exp(-a^2) + \frac{a}{2}(\text{erf}(a) + 1))^2$$

By numerical simulation we have $h'(a) \geq 0$, when $a \leq 0$, so $h(a)$ is increasing with $a$ when $a \leq 0$, thus

$$1 - \max(\frac{\mathbb{E}[\max(0, 1 - \mathcal{N}((\epsilon - \mu)\sqrt{d}, 1))]}{\mathbb{E}[\max(0, 1 + 1/\sqrt{2}(1 + \epsilon) - \mathcal{N}((\epsilon - \mu)\sqrt{\frac{d}{2}}, 0.5))]}, \frac{\mathbb{E}[\max(0, 1 - \mathcal{N}((\epsilon + \mu)\sqrt{d}, 1))]}{\mathbb{E}[\max(0, 1 + 1/\sqrt{2}(1 + \epsilon) - \mathcal{N}((\epsilon + \mu)\sqrt{\frac{d}{2}}, 0.5))]})$$

$$\geq 1 - h(-3) = 0.9775 > p,$$

by Theorem 8 we have the optimal solution $\mathbf{w}_t^* = (w_t^1, ..., w_t^{d+1})$ of our optimization problem must satisfy $w_t^1 \leq 1/\sqrt{2}$ and $w_t^2 = ... = w_t^{d+1}$ and $|w_t^2| \geq 1/\sqrt{2d}$. $\qquad\square$

**Lemma 11.** *When $w_t^1 \leq 1/\sqrt{2}$ and $w_t^2 = ... = w_t^{d+1}$ and $|w_t^2| \geq 1/\sqrt{2d}$, if $\epsilon_\infty \geq \frac{2}{d}\epsilon_1$ and $\epsilon_\infty \geq \sqrt{\frac{2}{d}}\epsilon_2$, $\infty$-player dominates 1-player and 2-player.. In another word, the training procedure cannot converge.*

*Proof.* Let $\ell_p = 1 - yw^\top(x + \boldsymbol{\delta}_p)$, we have

$$
\begin{aligned}
\ell_\infty - \ell_1 =& yw_t^\top(\boldsymbol{\delta}_1 - \boldsymbol{\delta}_\infty) = \epsilon_\infty\|w\|_1 - \epsilon_1\frac{\|\mathbf{w}_t\|_2^2}{\|\mathbf{w}_t\|_1} \\
&\geq \epsilon_1(\frac{2}{d}\|\mathbf{w}_t\|_1^2 - 1) \\
&\geq \epsilon_1(\frac{2}{d}(|w_t^1| + d|w_t^2|)^2 - 1) \\
&\geq \epsilon_1(\frac{2}{d}(\frac{1}{\sqrt{2}} + d\frac{1}{\sqrt{2d}})^2 - 1) > 0 \\
\ell_\infty - \ell_2 =& yw^\top(\boldsymbol{\delta}_2 - \boldsymbol{\delta}_\infty) = \epsilon_\infty\|w\|_1 - \epsilon_1\frac{\|w\|_2^2}{\|w\|_2} \\
&\geq \epsilon_2(\sqrt{\frac{2}{d}}\|\mathbf{w}_t\|_1 - 1) \\
&\geq \epsilon_2(\sqrt{\frac{2}{d}}(|w_t^1| + d|w_t^2|) - 1) \\
&\geq \epsilon_2(\sqrt{\frac{2}{d}}(\frac{1}{\sqrt{2}} + d\frac{1}{\sqrt{2d}}) - 1) > 0
\end{aligned}
$$

$$
\|w\|_1^2 - d\|w\|_2^2 = (w_1 + dw_2)^2 - dw_1^2 - dw_2^2 \geq (1-d)w_1^2 + d(d-1)w_2^2 \geq 0\,,
$$

Now, we have proved that $\infty$-player dominates others and $\text{sign}(w_i^t) = -\text{sign}(w_i^{t-1})$. With Lemma 8, we know that at any time $t$, we have $|w_t^i - w_{t-1}^i| \geq \sqrt{1/d}, \forall i \in [2, d+1]$, which means the training procedure cannot converge. $\square$

**Lemma 12.** *MAX and MSD are the same under the SVM scenario.*

*Proof.* Under the deep learning cases (non-linear and non-convex), MSD follows the steepest direction ($\ell_1$, $\ell_2$ or $\ell_\infty$) in each PGD step to find the perturbation which approximately maximizes the loss function, while MAX uses PGD to find the perturbations empirically and then chooses the perturbation maximizing the loss function. MSD and MAX are different approaches in deep learning cases (non-linear and non-convex).

On the other side, under the SVM (convex and linear) case, the optimal perturbations with $\ell_1$, $\ell_2$ and $\ell_\infty$ constraints have analytical solutions as shown in Eq. equation 2. In this way, both MSD and MAX can directly determine which perturbation maximizes the loss within one step, which means MSD and MAX are the same under the SVM case. $\square$

**Standard classification is easy.** Remind that the data consists of a robust feature $\mathbf{x}_1$, which is strongly related to the label and $d$ non-robust features $\mathbf{x}_i, i \in [2, d+1]$, which are weakly related to the label $y$. But with the non-robust features, we can construct a simple linear classifier $f$ that achieves over 99% natural accuracy as

$$
f(\mathbf{x}) = \text{sign}([0, \frac{1}{d}, \dots, \frac{1}{d}]^\top \mathbf{x})\,.
$$

For the natural accuracy, we have

$$
Pr[f(\mathbf{x}) = y] = Pr[\text{sign}([0, \frac{1}{d}, \dots, \frac{1}{d}]^\top\mathbf{x}) = y] = Pr[\frac{y}{d}\sum_{i=1}^{d}\mathcal{N}(\eta y, 1) > 0]
$$

$$
= Pr[\mathcal{N}(\eta, \frac{1}{d}) > 0] \geq 0.99\,,
$$

when $\eta \geq \frac{3}{\sqrt{d}}$.

**Robust classification is not easy**. We have the opposite observation when facing $\ell_\infty$ adversarial training. The robust accuracy is shown as

$$\min_{\|\boldsymbol{\delta}_\infty\|_\infty \leq \epsilon_\infty} Pr[f(\mathbf{x} + \boldsymbol{\delta}_\infty) = y] \leq Pr[\mathcal{N}(\eta, \frac{1}{d}) - \epsilon > 0] = Pr[\mathcal{N}(-\eta, \frac{1}{d}) > 0] \leq 0.01 \,,$$

when $\epsilon_\infty = 2\eta$. In the following part of our paper, we show that it is not only difficult to get a fairly good robust accuracy, but also a converged model under the multi-target adversarial training problem.

# D  Proofs

## D.1  Proof of Theorem 1

*Proof.* Combining Lemma 8, Lemma 11, and Lemma 12 yields this theorem. □

### D.1.1  Proof of Theorem 3

*Proof.* As the $\infty$-player dominates this game, the multi-target adversarial training problem reduces to the single-target problem equation 6. Further, with Lemma 8, for non-robust feature $i$, at any time $t$, we have $\text{sign}(w_i^t) = -\text{sign}(w_i^{t-1})$. Thus the training procedure does not converge. □

### D.1.2  Proof of Theorem 4

*Proof.* For the $i$-th player's loss (the $i$-th player dominates the bargaining game at the time $t$), as the loss function is $\mu$-strongly convex, we have

$$\ell_i(w^{t+1}) \geq \ell_i(w^t) - \ell_i'(w^t)^\top (w^{t+1} - w^t) + \frac{\mu}{2}\|w^{t+1} - w^t\|_2^2$$

$$\ell_i(w^{t+1}) \geq \ell_i(w^t) + \eta \ell_i'(w^t)^\top \ell_i'(w^t) + \frac{\mu\eta^2}{2}\|\ell_i'(w^t)\|_2^2 > \ell_i(w^t) \,.$$

For the $j$-th player's loss and $j \neq i$, as the loss function is $\mu$-strongly convex, we have

$$\ell_j(w^{t+1}) \geq \ell_j(w^t) - \ell_j'(w^t)^\top (w^{t+1} - w^t) + \frac{\mu}{2}\|w^{t+1} - w^t\|_2^2$$

$$\ell_j(w^{t+1}) \geq \ell_j(w^t) + \frac{\mu\eta^2}{2}\|\ell_i'(w^t)\|_2^2 > \ell_j(w^t) \,. \quad (\ell_i'(w^t)^\top \ell_j'(w^t) = 0, \forall i \neq j)$$

That means at time $t$, the loss of all player will keep increasing. And thus, if one player dominate the bargaining game throughout the whole game, the loss of all players and will keep increasing during the whole game, which means the bargaining game might not converge. □

### D.1.3  Proof of Theorem 5

*Proof.* As the loss function is $L$-smooth, $\forall i$, we have

$$\ell_i(w^{t+1}) \leq \ell_i(w^t) + \eta \ell_i(w^t)^\top (w^{t+1} - w^t) + \frac{L}{2}\|\eta \sum_{k \in [K]} g_k^t/K\|^2, \quad \text{(as L-smooth)}$$

$$\ell_i^{t+1} \leq \ell_i^t - \eta g_i^{t\top} \sum_{k \in [K]} g_k^t/K + \frac{L}{2}\|\eta \sum_{k \in [K]} g_k^t/K\|^2,$$

$$= \ell_i^t - \eta g_i^{t\top} g_i/K + \frac{L\eta^2}{2K^2} \sum_{k \in [K]} g_k^{t\top} g_k \,.$$

Summing the above inequality from $i = 1$ to $K$, we have

$$\ell^{t+1} \leq \quad \ell^t - \frac{\eta}{K} \sum_{k \in [K]} g_k^{t\top} g_k + \frac{L\eta^2}{2K} \sum_{k \in [K]} g_k^{t\top} g_k < \ell^t \,. \quad \text{(as } \eta < \frac{2}{L}) \tag{7}$$

The proof is completed. □

# E   Extra Experiments

Due to the limitation of space, we put the experimental verification of our negative results and Figure 4 here.

## E.1   Implementation Details

For each algorithm, we employ the default hyper-parameter introduced in their original papers. We implement all the methods on PyTorch (Paszke et al., 2019) with a single NVIDIA A100 GPU and optimized by the mini-batch SGD with the size of 128 and weight decay. The raw image is resized to $28 \times 28$ pixels for MNIST and $32 \times 32$ pixels for CIFAR-10 as inputs.

**Models.** Following Maini et al. (2020) and Madry et al. (2018), for MNIST, we use a four-layer convolutional network which consists of two convolutional layers of 32 and 64 $5 \times 5$ filters and 2 units of padding, followed by a fully connected layer with 1024 hidden units, where both convolutional layers are followed by $2 \times 2$ Max Pooling layers and ReLU activations. Similarly, following Maini et al. (2020), for CIFAR-10, we use the pre-activation version of the ResNet18 architecture that consists of nine residual units with two convolutional layers (He et al., 2016).

**Attacks used for training.** For MNIST, following Maini et al. (2020), the setting of three adversaries is shown below. The $\ell_\infty$ adversary uses a step size $\alpha = 0.01$ within a radius of $\epsilon = 0.3$ for 50 iterations. The $\ell_2$ adversary uses a step size of $\alpha = 0.1$ within a radius of $\epsilon = 2.0$ for 100 iterations, and the $\ell_1$ adversary uses a step size of $\alpha = 0.8$ within a radius of $\epsilon = 10$ for 50 iterations. By default, the attack is run with two restarts, one starting with $\delta = 0$ and another by randomly initializing $\delta$ in the allowable perturbation ball. Similarly, following Maini et al. (2020), for CIFAR-10, the setting of three adversaries is shown as follows. The $\ell_\infty$ adversary used a step size $\alpha = 0.003$ within a radius of $\epsilon = 0.03$ for 40 iterations. The $\ell_2$ adversary used a step size $\alpha = 0.05$ within a radius of $\epsilon = 0.5$ for 50 iterations, and the $\ell_1$ adversary used a step size $\alpha = 1.0$ with $\epsilon = 12$ for 50 iterations.

**Hyperparameter setting and tuning.** We did not tune any hyperparameters as our work is to show a phenomenon and solve it with our proposed adaptivebudget method. All the hyperparameters are directly copied from Maini et al. (2020). Specifically, on MNIST, for all the models, we used the Adam (Kingma & Ba, 2015) without weight decay, and used a variation of the learning rate schedule from Smith (2018), which is piecewise linear from 0 to $10^{-3}$ over the first 6 epochs, and down to 0 over the last 9 epochs. On CIFAR-10, for all the models, we used the SGD optimizer Robbins & Monro (1951) with momentum 0.9 and weight decay $5 \times 10^{-4}$. We used a variation of the learning rate schedule from Smith (2018) to achieve superconvergence in 50 epochs, which is piecewise linear from 0 to 0.1 over the first 20 epochs, down to 0.005 over the next 20 epochs, and finally back down to 0 in the last 10 epochs.

## E.2   Verifying effects of player domination on the SVM case

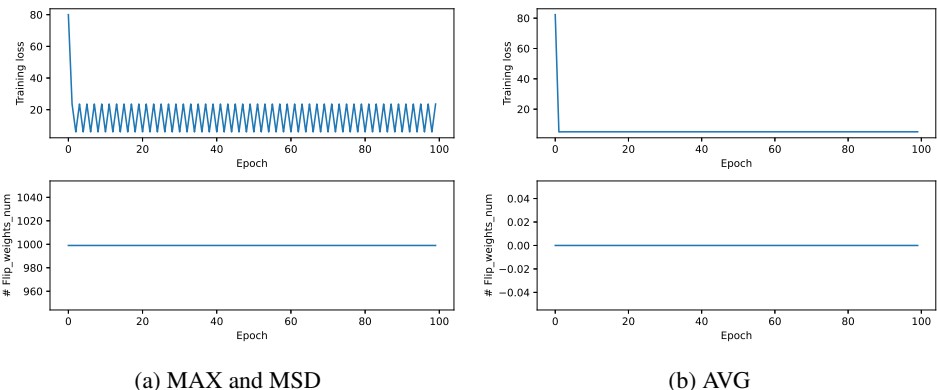

(a) MAX and MSD                              (b) AVG

Figure 1: We illustrate the training loss and the number of weights that flip between two epochs. Figure 1(a) shows the data of model trained with MAX and MSD using the SVM model (Sec 2.1) while Figure 1(b) shows the number of model trained with AVG.

To verify our theoretical results, we conduct experiments and the corresponding results are shown in Figure 1. We use a fully connected network (Fully Connected Layer (in=$d$, out=1), where $d = 1000$). For the data generation, we set $p = 0.95$, $\mu = 4/\sqrt{d}$, and $\epsilon_1 = \epsilon_2 = \epsilon_\infty = 2\mu$, and the sample size is 100000.

We notice that with MAX or MSD (they are equal under the SVM scenario, Lemma 12), the training procedure cannot converge as the training loss is fluctuating while the number of weights whose signs are flipped compared with last epoch is almost 1000. At the same time, AVG does converge. That complements our theoretical results (Theorem 1).

Besides, we also conduct experiments verifying the conjecture that when $\ell_1$ or $\ell_2$ player dominates the bargaining game, the training procedure does not converge as well. For the case when $\ell_1$ dominates, we set $\epsilon_1 = 4\mu, \epsilon_2 = \epsilon_\infty = 2\mu$, while when $\ell_2$ dominates, we set $\epsilon_2 = 4\mu, \epsilon_1 = \epsilon_\infty = 2\mu$. We observe exactly the same curves as Figure 1, showing that when $\ell_1$ and $\ell_2$ dominates, the training procedure with MAX and MSD cannot converge while with AVG, training procedure does converge. We present the results in Figures 2 and 3.

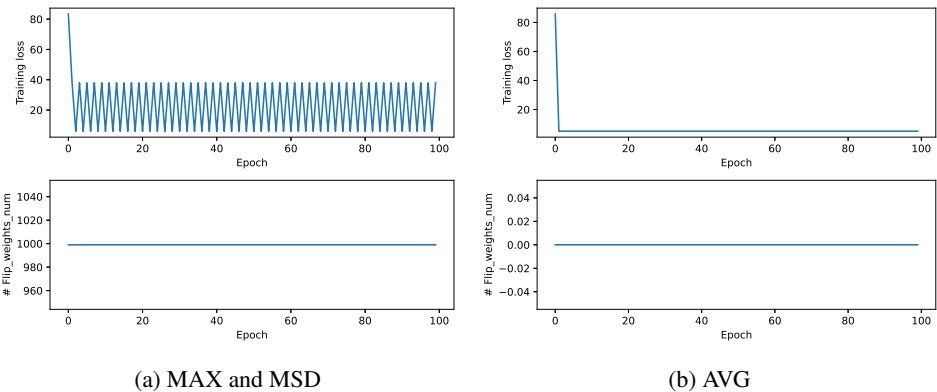

(a) MAX and MSD                    (b) AVG

Figure 2: We illustrate the training loss and the number of weights that flip between two epochs. We set $\epsilon_1 = 4\mu, \epsilon_2 = \epsilon_\infty = 2\mu$. Figure 2(a) shows the data of model trained with MAX and MSD using the SVM model (Sec 2.1) while Figure 2(b) shows the number of model trained with AVG.

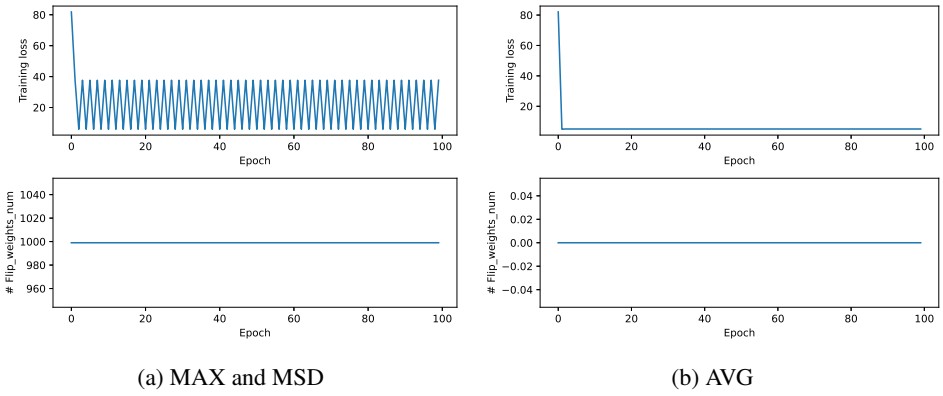

(a) MAX and MSD                    (b) AVG

Figure 3: We illustrate the training loss and the number of weights that flip between two epochs. We set $\epsilon_2 = 4\mu, \epsilon_1 = \epsilon_\infty = 2\mu$. Figure 3(a) shows the data of model trained with MAX and MSD using the SVM model (Sec 2.1) while Figure 3(b) shows the number of model trained with AVG.

**Norm choice in adaptive budget.** We use $\ell_1$ and $\ell_2$ norms for the adaptive budget and the corresponding results are shown in Table 1. There is no significant difference between the experiments with $\ell_1$ and $\ell_2$ norms when using our proposed method. The differences in overall robust accuracy are only $0.1\%$, $1.1\%$, and $1.0\%$ on MAX, MSD, and AVG respectively. The difference in each robust accuracy is also small.

## E.3  Results on CIFAR10

Similar, for the CIFAR10 dataset, we require that the adapted epsilon are bigger than the half of and smaller that twice of the original epsilon.

**Norm choice in adaptive budget.** We notice that the choice of norm in the adaptive budget barely influences the robust accuracy as shown in Table 2. On both three methods, *i.e.*, MAX, MSD, and AVG, our proposed adaptive budget is able to improve the performance with both $\ell_1$ and $\ell_2$ norms, while the difference between $\ell_1$ and $\ell_2$ norm is only 0.6% and 1.7%, 0.4% and 2.8%, 1.3% and 0.9% on MAX, MSD, and AVG against PGD and AutoAttack adversaries.

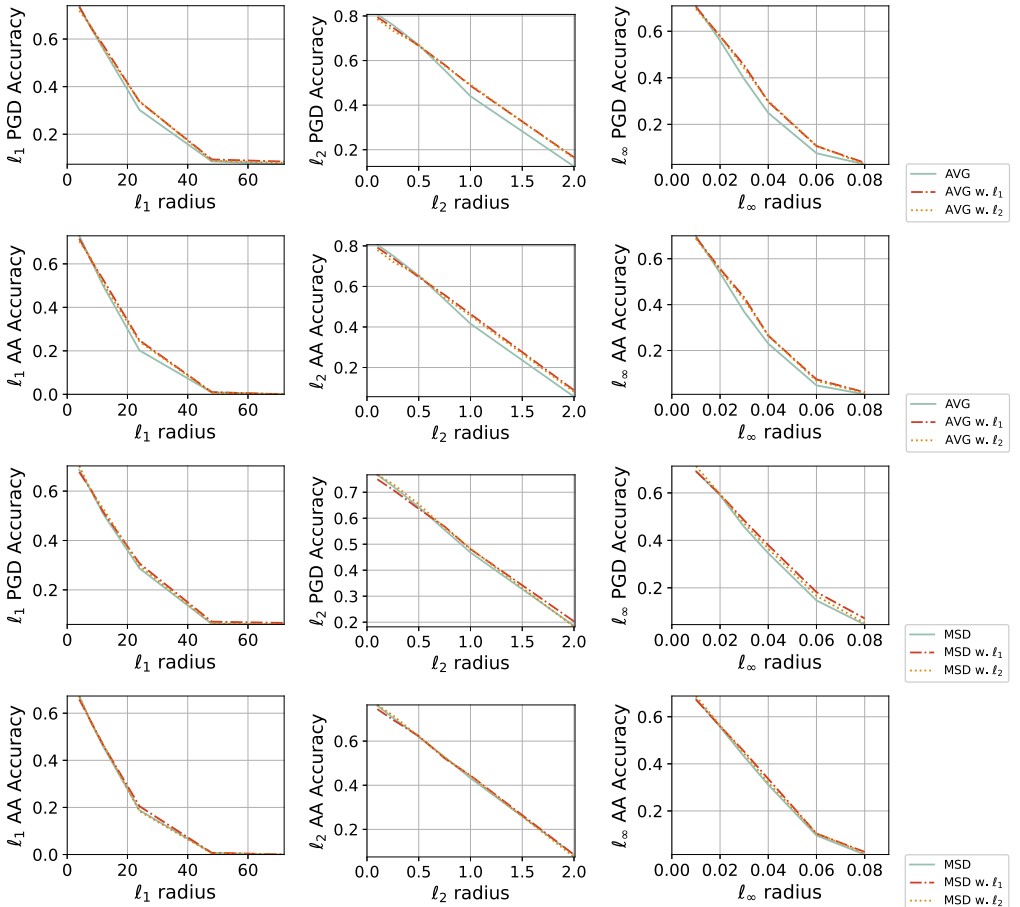

Figure 4: Robustness curves show the adversarial accuracy on CIFAR-10 trained with MSD and AVG against $\ell_1$ (left), $\ell_2$ (middle), and $\ell_\infty$ (right) perturbation models over a range of epsilon.

**Robustness curves.** The robustness curves are shown in Figures 5 and 4. The lines of MAX with either $\ell_1$ or $\ell_2$ norm-based adaptive budget method are higher than the lines without the adaptive budget method. The gap between lines with the adaptive budget method and lines without is biggest when the budget of the adversary is small. Similar observation can be obtained from the line of MSD and AVG in Figure 4.

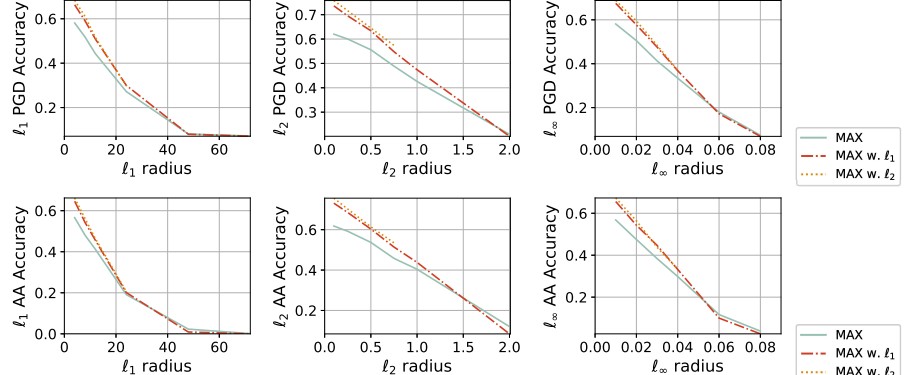

Figure 5: Robustness curves show the adversarial accuracy on CIFAR-10 trained with MAX against $\ell_1$ (left), $\ell_2$ (middle), and $\ell_\infty$ (right) PGA and AutoAttack ("AA" in the figures) perturbation models over a range of epsilon. Plots of MSD and AVG are similar and thus deferred to Appendix.

