# OpenReview forum: "Cooperation or Competition: Avoiding Player Domination for Multi-target Robustness by Adaptive Budgets"
_NeurIPS.cc/2022/Workshop/TSRML — TSRML2022_

### Official Review · Reviewer_CVwH · 2022-10-18

**Overall Recommendation:** A good paper with interesting insight…
**Overall Rating:** 7

**Summary:**

This paper proposes an adaptive budget strategy for multi-target adversarial training. Empirical evaluation is done on MNIST and CIFAR-10, showing promising improvements.

**Strengths:**

The idea of adaptively tuning the budgets in multi-target adversarial training is interesting, and the algorithm is simple to implement. Empirical evaluation is done under strong attacks like AutoAttack, achieving promising results.

**Weaknesses:**

Investigating the effect of more general $\ell_{p}$-norm beyond $\{1,2,\infty\}$ would be more intriguing.

**Review Confidence:**

5: The reviewer is absolutely certain that the evaluation is correct and very familiar with the relevant literature

---

### Official Review · Reviewer_h44o · 2022-10-18

**Overall Rating:** 6

**Summary:**

The authors propose to cast multi-targeted (against multiple attack types) adversarial training as a cooperative bargaining game. They then proceed to show that under certain circumstances a MAX or MSD aggregation of gradients leads to a phenomenon they coin "player domination", describing the situation where a single attack (type) dominates (the gradients of) all others. This setting can lead to non-convergence and poor performance against a joint adversary. To tackle this challenge they propose AdaptiveBudget, an algorithm that adaptively changes the budget available to the different attackers to avoid the player domination phenomenon. They demonstrate the effectiveness of their method in an empirical study.

**Strengths:**

* The paper addresses the important problem of multi-targeted robustness by providing a method that is orthogonal to other approaches.
* Interesting theoretical analysis based on viewing multi-targeted training as cooperative bargaining.
* The paper is easy to follow although the presentation could be improved in some sections (e.g. redundancy between L141ff and L137ff (and both theorems)).
* Convincing empirical results, leading to an improved "All PGD" and "All AA" robustness in all settings.

**Weaknesses:**

* The authors do not demonstrate theoretically that their approach prevents player domination.
* A more thorough theoretical analysis showing the distribution of perturbation budgets and a comparison of training with the mean budgets (per attack type) would be interesting to assess the importance of the adaptiveness of the proposed approach, versus a suitable choice of relative perturbation magnitudes.
* While convergence under player domination can theoretically be problematic, this corresponds to training for a single adversary, which is a well-studied problem and does practically not suffer from non-convergence issues, weakening the theoretical motivation.
* Without careful inspection of the algorithm, it is not clear that the attack budget is in fact "reset" after every step and changes do not stack. I would suggest improving the clarity of the corresponding section

### Minor Comments
* Unclear which inner maximization problem is referred to in L89. Norm-wise optimal perturbations do not seem to depend on the aggregation mode.
* The boldening scheme in Table 1 seems non-standard (highlighting exactly the improvements of the own method). Consider boldening the best result per row or per row and category.


**Overall Recommendation:**

The proposed method is orthogonal to other advances in training for multi-targeted robustness and empirically improves performance against the worst-case attack across all settings. While casting this training problem as cooperative bargaining provides interesting insights, the connection to the proposed AdaptiveBudget algorithm seems not sufficiently substantiated (either theoretically or empirically). Further, while the empirical improvements over the baseline methods are convincing, the connection to the theoretical motivation seems weak. Overall I believe the merits marginally outweigh the flaws but encourage the authors to provide additional empirical and theoretical results connecting their theoretical contributions with the Adaptive Budget algorithm.

**Review Confidence:**

3: The reviewer is fairly confident that the evaluation is correct

---

### Decision · Program_Chairs · 2022-10-23

**Decision:**

Accept

**Comment:**

Following the unanimous recommendations from reviewers, the submission is accepted.